# Subclinical Reactive Hypoglycemia with Low Glucose Effectiveness—Why We Cannot Stop Snacking despite Gaining Weight

**DOI:** 10.3390/metabo13060754

**Published:** 2023-06-15

**Authors:** Ichiro Kishimoto

**Affiliations:** Department of Endocrinology and Diabetes, Toyooka Public Hospital, 1094, Tobera, Toyooka 668-8501, Hyogo, Japan; ichirou-kishimoto@toyookahp-kumiai.or.jp; Tel.: +81-796-22-6111

**Keywords:** subclinical reactive hypoglycemia, appetite, glucose effectiveness, obesity, overweight, snacking habits

## Abstract

Obesity has grown worldwide owing to modern obesogenic lifestyles, including frequent snacking. Recently, we studied continuous glucose monitoring in obese/overweight men without diabetes and found that half of them exhibit glucose levels less than 70 mg/dL after a 75-g oral glucose load without notable hypoglycemic symptoms. Interestingly, people with “subclinical reactive hypoglycemia (SRH)” snack more frequently than those without it. Since the ingestion of sugary snacks or drinks could further induce SRH, a vicious cycle of “Snacking begets snacking via SRH” can be formed. Glucose effectiveness (Sg) is an insulin-independent mechanism that contributes to most of the whole-body glucose disposal after an oral glucose load in people without diabetes. Our recent data suggest that both higher and lower Sg are associated with SRH, while the latter but not the former is linked to snacking habits, obesity, and dysglycemia. The present review addresses the possible role of SRH in snacking habits in people with obesity/overweight, taking Sg into account. It is concluded that, for those with low Sg, SRH can be regarded as a link between snacking and obesity. Prevention of SRH by raising Sg might be key to controlling snacking habits and body weight.

## 1. Introduction

Obesity, or being overweight, has grown epidemiologically, with over 4 million people dying each year as a result of the global burden. To make matters worse, rates of overweight and obesity are rising further, both in adults and children. In fact, in children and adolescents, the prevalence of overweight or obesity increased more than 4-fold from 4% to 18% globally during the recent 40 years [1]. This is thought to be due to an obesogenic environment in modern lifestyles (cheap calorie-dense food, social networking or television images that portray food in a very appetizing way (“Meshi tero” in Japanese), communities that lead to lower physical activity, and increases in nonphysical entertainment) [2]. In addition, excessive emphasis on low-fat intake has resulted in excessive intake of simple carbohydrates, such as sugary snacks.

The growing trend for snacking, which contributes to excess energy intake and weight gain, could be one of the leading causes of the present obesity pandemic seen worldwide [3]. To prevent obesity and obesity-related morbidities, including diabetes and atherosclerotic cardiovascular diseases, snacking habits may be an important lifestyle component [4], and a reduction of the number of calories/carbohydrates consumed from snacks could lower the risk of overweight and obesity. However, since snacking has been so incorporated into the modern lifestyle that eating habits are shifting from the traditional three meals per day to more frequent snacking, it is not easy to stop or reduce snacking. In addition, since the main location of snacking is at home [5], where many people are stuck during the COVID-19 pandemic, the trend has intensified in the past few years [6].

Recently, we reported that, in obese/overweight subjects without diabetes, half exhibit blood glucose levels less than 70 mg/dL without notable hypoglycemic symptoms except hunger during continuous glucose monitoring within 24 h after a 75-g oral glucose challenge (OGTT) [7]. Since “subclinical” reactive hypoglycemia (SRH) was significantly associated with snacking frequency during the 6 days after a glucose load, we propose that, even subclinical, SRH is linked to snacking behavior in obese/overweight subjects, possibly to prevent overt hypoglycemia. Since preventing hypoglycemia is a matter of life and death, snacking on SRH is not an easily modifiable factor. The present review addresses the possible role of SRH in the habitual ingestion of sugary snacks in people with obesity or overweight, taking glucose effectiveness (Sg), an insulin-independent mechanism of blood glucose disposal, into account. Although being obese or overweight is often stigmatized as greed and a lack of willpower, it is suggested that snacking in certain conditions may not be self-controllable.

## 2. Snacking and Obesity

Although other factors may also be involved, the primary cause of adiposity in obesity/overweight is an imbalance of calories eaten and burned. As there is a tendency worldwide toward an increase in the intake of food that is high in fat and sugar [8], the substantial increase in snacks’ total energy and energy density may contribute to the obesity epidemic. Several epidemiological studies have suggested an association between snacking and obesity [9,10]. A meta-analysis of observational studies in children/adolescents addressing the associations between overweight/obesity and dietary/behavioral risk factors revealed that the odds ratio (95% confidence interval (CI)) associated with childhood overweight/obesity for drinking sugar-sweetened beverages ≥4 times/week was 1.24 (1.07, 1.43) [11]. Subjects with obesity or overweight consume more snacks daily and more calories at each snacking occasion compared to those with normal weight [12].

On the contrary, studies showed that snackers were less likely to be overweight or obese and less likely to have abdominal obesity [13,14,15]. The odds ratio associated with childhood overweight/obesity for eating snacks ≥4 times/week was 0.84 (CI 0.71, 1.00) [11]. The reasons for the discrepancy among study results on snacking and its association with obesity remain to be clarified, but could include that the size and type of snacks may differ between studies and that factors such as the amount, quality, and pattern of diets other than snacks or of exercise/physical activity are not well controlled. Although, in theory, regular ingestion of snacks in large portions to exceed the body’s daily energy needs could contribute to weight gain, as of today, researchers have failed to present a definite positive causal relationship between snacking frequency and long-term weight gain in a prospective study [16]. The results of the previous epidemiological studies indicate that excess calories from frequent snacking may be associated with but not sufficient for developing obesity, where additional factor(s) may also be involved.

## 3. Blood Glucose and Appetite

Although peptidic signalings such as leptin, ghrelin, glucagon-like peptide (GLP)-1, and so on have been in the spotlight in recent appetite research, blood glucose is also involved in appetite expression, as was originally proposed by Jean Mayer in his ‘glucostatic hypothesis’ [17], where meal initiation is triggered by a fall in blood glucose as feeding is terminated by a rise in it. It seems pertinent that the brain regulates appetite by detecting plasma glucose levels since glucose plays a key role in the energy delivery of the central nervous system. Since most of the energy supply to the cerebral demand is delivered by glucose, it is reasonable that a reduction in glucose is acutely sensed by the hypothalamic brain areas and activates the several counter-regulatory systems against hypoglycemia, aiming to avoid neuroglycopenia, whose symptoms include loss of consciousness, confusion, personality change, or bizarre behavior. Indeed, in people with diabetes medications, low glucose levels prompt them to consume carbohydrates to combat intense hunger [18], which can be regarded as a warning sign to prevent neuroglycopenia. Although the concept of the “glucostat theory” was supported by animal studies with glucose deprivation and by clinical observations with symptomatic hypoglycemia, the idea of glucose as a primary metabolic signal in humans remained inconsistent since fasting glucose concentration is usually tightly regulated in healthy subjects [19].

Conventional glucose measurements, such as those with venipuncture or finger pricks, do not capture precise fluctuations in blood glucose over time, i.e., glucose variability. Continuous glucose monitoring (CGM) is wearable technology that, through a tiny sensor inserted subcutaneously, automatically tracks blood glucose levels continuously 24 h a day. Recently, CGM was introduced in daily clinical practice, and the glucose fluctuation data in daily living detected by CGM are used to evaluate changes in behavior or medications, facilitating shared decision-making among people with diabetes and their healthcare providers [20]. However, studies using CGM in people without diabetes have been scarce, and glucose variability in seemingly healthy people has not been fully studied. Recently, Kim et al. examined the CGM of healthy nondiabetic individuals in a free-living setting and found that glucose nadirs before a meal predicted subsequent hunger and food intake [21]. In addition, using CGM in their prospective cohorts, Wyatt et al. showed that the average postprandial glucose dips 2–3 h after a standardized meal (890 kcal, 86 g of carbohydrates, 53 g of fat, 16 g of protein, and 2 g of fiber, ingested within 10 min) relative to baseline level are significantly associated with an increase in hunger postprandial for 2–3 h, a shorter time until the next meal, and greater postprandial energy intake [22]. Therefore, lower blood glucose levels may perform crucial functions in the regulation of appetite and energy intake not only in the clinical setting but also in daily life.

## 4. Reactive Hypoglycemia and Snacking Frequency in Obese/Overweight

“Reactive hypoglycemia” has been termed a condition characterized by recurrent episodes of typical hypoglycemia symptoms with documented low glucose (<70 mg/dL) occurring within 2–4 h after a high carbohydrate meal in otherwise healthy individuals [23]. While periodical testing of ambulatory blood glucose by finger prick, which has been used to detect reactive hypoglycemia, provides only snapshots of postprandial blood glucose levels, utilization of CGM could provide a practical means for confirming reactive hypoglycemia, facilitating early intervention [24].

We have previously conducted CGM in obese/overweight participants for 6 days after 75-g OGTT and reported that, in non-diabetes, most of whom exhibit normal glucose tolerance on OGTT, 47% had CGM-recorded sensor glucose levels of ≥200 mg/dL at least once, whereas approximately 30% had CGM glucose levels ≥180 mg/dL at least once in every 5 meals in their daily lives [25]. In addition, lifestyle factors such as snacking and physical inactivity served as the major drivers of postprandial hyperglycemia independent of insulin-related indices [26]. During the study, we also found that in half of the subjects, the minimal sensor glucose levels within 24 h post-OGTT were less than 70 mg/dL, the clinical threshold of hypoglycemia, while no symptom except hunger was observed in one participant [27]. When the relationship between the SRH and times of snacking during 6 days after a glucose challenge was examined, the median snacking frequency was 6 times higher in subjects with SRH compared to those without it (0.9 vs. 0.15 times/day) (*p* < 0.001) [7]. It is, therefore, suggested that, even without a notable symptom, mild hypoglycemia reactive to a glucose load is significantly associated with higher snacking frequency in obese/overweight males without diabetes.

## 5. Effect of SRH on the Relationship between Snacking Habits and Obesity

In the obese/overweight subjects, we next examined if habitual snacking is related to obesity. Among participants with CGM sensor recordings of ≥720 (60 h) (*n* = 43), there was no difference in body mass index (BMI) in subjects with self-reported snacking habits compared to subjects without snacking habits (median BMI 27.7 vs. 27.9, *p* = 0.9). Nonetheless, the proportion of obesity (BMI ≥ 30) was significantly higher in subjects with snacking habits compared to those without snacking habits (35.7% vs. 13.8%, *p* = 0.049). When subjects were stratified by the presence or absence of SRH, almost half (46%) of those with snacking habits were obese (BMI ≥ 30) in the presence of SRH, while none of those exhibited obesity in the absence of SRH [7]. When the relationships between snacking frequencies (snacking days per week) and BMI (kg/m^2^) were fit with spline curves according to the presence or absence of SRH, less snacking was observed by increasing BMI in those without SRH, while in subjects with SRH, obese people still snack regularly even if their BMI is high (Figure 1). When the subjects were divided by the median BMI (27.9 kg/m^2^), the proportions of self-reported snacking habits (≥once/week) in the lower and higher BMI groups were 35.7% and 46.2% in the presence of SRH, whereas those were 42.9% vs. 0% (Fisher’s exact *p* = 0.015) in the absence of SRH (calculated from data presented in [7]). It is, therefore, suggested that, while people tend to stop snacking habits when their body weight increases, those who have SRH cannot control habitual snacking, possibly because of instinctive prevention of hypoglycemia.

Plots of BMI (kg/m^2^) against snacking frequency (days per week) were created for 43 males from data previously reported [7]. Data were further assessed by the cubic spline functions and smooth curve fitting with lambda = 1 (the solid line in A and the dashed line in B). SRH was defined as previously reported [7]. Snacking frequencies were self-reported. BMI was defined as body weight (kg) divided by height (m) squared. The open circles denote each person without SRH, while the closed circles denote each person with SRH.

## 6. Biphasic Effect of Glucose Effectiveness on Reactive Hypoglycemia

The term “glucose effectiveness (Sg)” has been applied to describe the sum of the insulin-independent regulatory mechanisms of glucose disposal and appearance, where glucose increases its utilization in the periphery and inhibits its production in the liver. Sg is one of the important systems for the maintenance of normal blood glucose levels. The contribution of Sg to whole-body glucose disposal after an oral glucose load is estimated to be ~50% in healthy people, ~85% in individuals who are overweight, and ~99% in individuals with severe insulin resistance [28]. Independently of insulin action, doubling plasma glucose results in ~50% suppression of hepatic glucose production (HGP) [28], with reductions in the rates of both gluconeogenesis [29] and glycogenolysis [30]. It is reported that Sg-mediated suppression of HGP was impaired in the presence of elevated plasma free fatty acid concentrations [31]. To estimate Sg, either a euglycemic pancreatic clamp or a frequently sampled intravenous glucose tolerance test (FSIVGTT) is required [32], both of which are complicated and laborious. To overcome this issue, Nagasaka et al. describe an index of Sg derived from a standard 75-g OGTT and name the index “oral Sg index (SgIo)” [33].

In our previous study, in obese/overweight males without a diagnosis of diabetes (*n* = 43), SRH was significantly associated with the tertile category of SgIo in a biphasic manner [34]. The proportions of SRH were 40.7%, 18.5%, and 40.7% for the lowest, middle, and highest SgIo tertiles, respectively (Chi-square *p* = 0.014). In addition, the proportions of self-reported snacking habits are significantly associated with the SgIo tertile categories (57.4%, 14.3%, and 28.6% for the lowest, middle, and highest SgIo tertiles, respectively) (Chi-square *p* = 0.037). The odds ratios for having SRH in the lowest and highest SgIo tertiles compared to those in the middle SgIo tertile were both 7.3 (CI 1.4–38.9, one-sided Fisher *p* = 0.018), while the odds ratios for having snacking habits in the lowest and highest SgIo tertiles compared to those in the middle SgIo tertile were 8.7 (CI 1.4–53.8, *p* = 0.017) and 2.6 (CI 0.4–17.2, *p* = 0.291), respectively. The odds of having obesity (BMI ≥ 30) and impaired glucose tolerance (blood glucose levels at 2 h after a 75-g glucose load ≥ 140 mg/dL) were significantly higher in subjects with the lowest SgIo tertile category compared to the middle category, with odds ratios of 14.0 (CI 1.4–137.3, *p* = 0.013) and 11.7 (CI 1.8–74.2, *p* = 0.007), respectively, while there were no significant differences between the highest and middle SgIo tertiles. It is, therefore, suggested that both higher and lower Sg are associated with SRH, while the latter but not the former is linked to snacking habits, obesity, and dysglycemia.

## 7. SRH in Subjects with Higher and Lower Sg

Since higher Sg leads to higher glucose disposal, an increase in Sg could play a critical role in hypoglycemia reactive to an oral glucose load. In our previous study, SRH was significantly correlated with higher eating/snacking frequencies [7]. Thus, higher eating frequency in subjects with higher SgIo can be considered a protective mechanism against symptomatic hypoglycemia. On the other hand, subjects with lower SgIo exhibited postprandial hyperglycemia [27]. In addition, post-75-g glucose challenge hyperinsulinemia is associated with the lower SgIo category, indicating hypoglycemia in these subjects is dependent on insulin excess in response to hyperglycemia. The hypoglycemia-induced increase in appetite could lead to excess caloric intake through unhealthy sugary snacking, thus forming a vicious cycle of snacking, postprandial hyperglycemia, delayed hyperinsulinemia, SRH, and an increase in appetite. Lower SgIo was tightly associated with higher BMI, impaired glucose tolerance, and insulin resistance [34]. Therefore, for obese/overweight subjects with low Sg, SRH can be regarded as a link between snacking and obesity, and preventing SRH might be key to controlling body weight in these people.

## 8. Lifestyles Related to Sg

To date, several lifestyle factors have been reported to be associated with Sg. Exercise training improves Sg [28,35,36,37], possibly through GLUT4 translocation to the plasma membrane [38] and through AMP-activated protein kinase (AMPK) [39]. In our recent study, in obese/overweight men without diabetes, lower SgIo is associated with a lower percentage of days when subjects walk ≥8000 steps/day, while walking habits (≥8000 daily steps, ≥60% of days) were associated with higher SgIo, delineating the role of light daily exercise such as walking on Sg [27]. In addition, high-fat diets are reported to reduce the contribution of Sg to glucose disposal [40]. Furthermore, the exercise-induced enhancement of Sg is suppressed by a high glycemic index diet, possibly through plasma free fatty acids [41]. Frequent ingestion of instant noodles, whose average content of saturated fatty acids is considerably high among cereal foods, showed a tendency toward lower SgIo [27]. Taken together, Sg is reduced in subjects with high-fat diets or sedentary daily lives, as often seen in modern lifestyles. On the contrary, it is high in those with low-fat diets or physically active daily lives.

## 9. Historical Perspectives on SRH-Induced Appetite and Glucose Effectiveness

Before the Industrial Revolution, our ancestors included strenuous physical activity as a normal part of their daily lives [42]. Physical activities were conducted as part of standard work either inside or outside the home. In clear contrast, modern lives have become increasingly sedentary through increased use of motorized transport as well as screens for work, home, school, and recreation. The development of the internet and its availability on mobile devices have also spurred physical inactivity [43]. In addition to decreased physical activity, today’s “obesogenic environment” includes easy, 24 h access to energy-dense processed foods, particularly those high in fat and sugar [44]. Therefore, the quantity and quality of exercise, as well as dietary fat, have been subject to tremendous change over the past half-century, causing a conflict with our genome, which can adapt more slowly [45].

Although, in the low fat/high exercise condition as seen in pre-modern lives, an increase in adiposity and a decrease in Sg are protective mechanisms against SRH, in the high fat/low exercise condition as seen in present westernized lives, it promotes post-prandial hyperglycemia and hyperinsulinemia, which augment SRH and no longer serve its original purpose (Figure 2). In a society where snacking and motorization predominate, people with low Sg cannot help eating sugary snacks regularly to prevent SRH, while sugary snacks rich in fat induce further SRH, thus forming a vicious cycle of “Snacking begets snacking”, which leads to the establishment of snacking habits. Since “Fighting the biological urge to eat is a losing battle by itself” [46], it is extremely difficult to combat the SRH-induced appetite, which is like going against nature. Therefore, in obese people with low Sg, prevention of SRH by raising Sg is mandatory to stop snacking habits and control body weight.

The vertical axis shows the likelihood of the respective parameters (i.e., weight gain, post-meal hyperglycemia, hyperinsulinemia, insulin resistance, increase in appetite, and SRH), while the horizontal axis shows the level of Sg.

(Left side) In the low-fat/high-exercise condition as seen in pre-modern lives, glucose effectiveness, an insulin-independent mechanism of glucose disposal, becomes high, which is associated with frequent mild hypoglycemia without typical symptoms (thus, subclinical) after meals. The subclinical hypoglycemia-induced increase in appetite is regarded as a protective mechanism against symptomatic hypoglycemia.

(Right side) In the high-fat/low-exercise condition as seen in modern lives, glucose effectiveness becomes low, which is associated with frequent postprandial hyperglycemia and hyperinsulinemia. In people at higher risk of developing diabetes, such as those with overweight/obesity, defective early insulin secretion and delayed insulin responses lead to frequent postprandial subclinical hypoglycemia. Upon global availability of energy-dense snack foods and beverages, the subclinical hypoglycemia-induced increase in appetite likely leads to frequent ingestion of snacks rich in fat and sugar, which increases the frequencies of post-prandial hyperglycemia and hyperinsulinemia, creating a vicious cycle of the establishment and maintenance of obesity.

## 10. Childhood Lifestyles to Maintain Sg High

The United States state-based telephone interview survey conducted by the Centers for Disease Control and Prevention and state health departments showed that, in adults, the South had the highest prevalence of obesity (BMI of 30 kg/m^2^ or higher), followed by the Midwest, the Northeast, and the West [47]. Although the percentage of people with obesity has been soaring during the last decade (the United States obesity prevalence increased from 30.5% in 1999–2000 to 41.9% in 2017–2020 [48]), regional differences in the prevalence of obesity are maintained [49]. Amador et al. explored causes of regional variation in data on obesity-related traits as indicators of the health status of ~11,000 Scottish individuals with genotypic records and a variety of measurements of possible causal lifestyle and socioeconomic factors, where it is suggested that regional variation for most obesity traits was associated with lifestyle and socioeconomic variables, such as smoking, diet, and deprivation, which are potentially modifiable [50], while the regional differences in obesity exist even after accounting for genetics [50].

In addition, the United States regions with the highest prevalence of obesity (BMI > 95th percentile) in children roughly correspond with those in adults [51]. Although race and ethnicity among regions may account for the disparity among regions, a similar trend can be observed in Japan [52], a country characterized by much less racial heterogeneity compared to the United States. The regional agreement on obesity prevalence among children and adults may suggest sharing and inheriting similar psychosocial, economic, or familial environments, which strongly influence lifestyles that result in or prevent obesity in both adults and children. According to a survey of 2000 Americans, half of them believe they have the same snacking habits as their parents [53], and 60% say their snacking habits have ties to their cultural heritage, indicating that snacking habits are one of the major obesity-related lifestyles shared in the family. SRH leads to frequent ingestion of sugary snacks or drinks since people who face life-threatening situations such as hypoglycemia learn to avoid them. Oftentimes, they study how to behave correctly, such as “a burnt child dreads the fire”. However, if Sg is low, the rescue behavior ironically augments the unfavorable situation, and, thus, snacking habits are formed, a negative legacy passed for generations from parents to children.

In Japan, from 1975 to 2000, the rates of childhood obesity grew from 6% to 12% among boys and from 6% to 10% among girls [52]. Although it nearly doubled, the increases are less than those in other countries, such as the United States, where the prevalence of childhood obesity has tripled over that period. Furthermore, since 2000, the frequency of overweight/obese children has declined slightly for both boys and girls. The reason the rates seem to stay exceptionally low in Japan is of great interest to other countries, which see their children becoming less and less healthy in terms of obesity. It is speculatively attributed to Japan’s school policies, including well-balanced school lunches [54] and walking to/from school (instead of using a school bus or car) [55]. Given that healthy eating and physical activity increase Sg, regional obesity inequalities can be tackled with appropriate interventions from childhood, taking Sg into account.

## 11. Possible Pharmacological Intervention

Given that SRH induces appetite, pharmacological prevention of SRH may become a treatment for obesity-related diseases, although the beneficial effects of any interventions should be balanced with their adverse reactions and costs. Glucagon is a pancreatic hormone that contributes to maintaining normal glycemia by activating glycogenolysis and gluconeogenesis in the liver. In humans, suppression of plasma glucagon concentrations was observed with a nadir at 40 min following OGTT [56], where vagal transmission may be involved [57]. Since glucagon, which is utilized as a treatment for clinical hypoglycemia, raises blood glucose levels, supplementation with glucagon may prevent SRH.

Glucose-dependent insulinotropic polypeptide (GIP) is an intestinally derived peptide secreted from the upper small intestine in response to feeding. The GIP receptor is expressed in pancreatic α- and β-cells, indicating that it controls postprandial glucose levels [58]. Several studies showed that infusion of GIP counteracts glucose-induced suppression of glucagon secretion [59]. Therefore, supplementation with exogenous GIP may also prevent SRH. Recent findings in type 2 diabetes show that, when paired with GLP-1, GIP receptor agonist therapy produces profound weight loss and glycemic control [60]. Tirzepatide, a once-weekly subcutaneous injectable peptide with agonist activity at both the GIP and GLP-1 receptors [61], provided substantial and sustained reductions in body weight [62]. Although post-glucose challenge suppression of glucagon is reduced in people with type 2 diabetes [56], upregulation of glucagon and inhibition of SRH-induced appetite may be one of the mechanisms of tirzepatide’s action in reducing body weight.

In addition, as shown in the following paragraphs, brown adipocytes and low-grade inflammation could also be potential therapeutic targets in the future.

## 12. Brown Adipose Tissue (BAT) and Sg

In healthy women treated with a β3-adrenergic receptor agonist, it is reported that the metabolic activity of BAT measured by [18F]-2-fluoro-d-2-deoxy-d-glucose (18F-FDG) PET/CT is increased, which is accompanied by higher Sg as well as insulin sensitivity and insulin secretion [63], suggesting that glucose uptake by BAT may increase Sg.

Cold exposure is one of the best-explored activators of brown adipocytes. During the ice age, the latest of which peaked about 20,000 years ago, it is speculated that global temperatures were likely about 10 °F (5 °C) colder than today [64]. Global warming as well as heating equipment such as air conditioning in civilized countries may lead to the deactivation of brown adipocytes, which could also be involved in the tendency towards low Sg in modern lives. Since BAT increases not only energy expenditure but also glucose clearance [65], earlier loss of brown adipocytes could lead to decreased postprandial glucose disposal and, thus, to hyperglycemia after meals.

## 13. Postprandial Inflammatory Response and Sg

It is known that ingestion of a high-fat diet, high-carbohydrate diet, or combination causes postprandial responses associated with activation of the innate immune system, which initiates systemic low-grade inflammation [66]. High-fat meal consumption results in a temporary pro-inflammatory state via “metabolic endotoxemia”, in which bacterial wall products derived from gut microbiota such as lipopolysaccharide (LPS) can be found in the blood circulation [67]. In healthy volunteers, endotoxin challenge (1 ng/kg LPS) significantly decreased Sg derived from a frequently sampled intravenous glucose tolerance test [68]. Therefore, increases in plasma LPS and pro-inflammatory cytokines such as interleukin-6 (IL-6) or tumor necrosis factor-α (TNF-α) levels could lead not only to insulin resistance [67] but also to decreased Sg [68], both of which may result in postprandial hyperglycemia and subsequent reactive hypoglycemia.

In mice models with gut microbiota depletion, it is reported that glucose uptake in BAT is enhanced [69]. A local pro-inflammatory environment in BAT alters its thermogenic activity by impairing its energy expenditure mechanism and glucose uptake for use as a fuel substrate [70]. Therefore, “metabolic endotoxemia” might lead to the inhibition of glucose uptake by BAT, which results in lower energy expenditure as well as postprandial hyperglycemia, especially when the amount of active BAT is abundant, such as in children and adolescents or in adults with cold exposure.

## 14. Reactive Hypoglycemia and Inflammation

Previous studies documented that, in both diabetes and non-diabetes, markers of inflammation such as C-reactive protein and IL-6 are upregulated after acute hypoglycemia [71,72,73]. Combined with endothelial dysfunction, hypercoagulability, and activation of the sympathetic nervous system, the chronic inflammatory mechanism is thought to be involved in the pathogenesis of atherosclerotic diseases in postprandial reactive hypoglycemia [74,75].

It is reported that several pro-inflammatory cytokines such as TNF-α, interleukin (IL)-6, IL-1α, and IL-1β can cross the blood-brain barrier (BBB), while receptors for these pro-inflammatory cytokines are expressed in the hypothalamus [76]. The pro-inflammatory cytokines-induced low-grade inflammation changes the levels of neuropeptides involved in the hypothalamic and mesolimbic appetite-controlling systems, which could be associated with overeating through non-homeostatic (hedonic) pathways [77]. Along with nutritional signals such as saturated fatty acids, which have been shown to cause chronic hypothalamic inflammation, an increase in food intake, and weight gain [78], chronic low-grade inflammation caused by SRH might also contribute to excess food intake.

## 15. Conclusion and Future Directions

In conclusion, although the SRH-induced increase in appetite can be regarded as a protective mechanism to maintain normal glucose levels and prevent neuroglycopenia in high Sg conditions, it may create a vicious cycle that leads to obesity in low Sg conditions. Accompanied by modern environmental changes, Sg could become increasingly lower, which would shift SRH-induced appetite from an originally compensatory mechanism to a strong driver of obesity and possibly “typically Western” diseases. In obese patients with low Sg, treatment or prevention of obesity should focus on how to raise Sg and avoid SRH.

## Figures and Tables

**Figure 1 metabolites-13-00754-f001:**
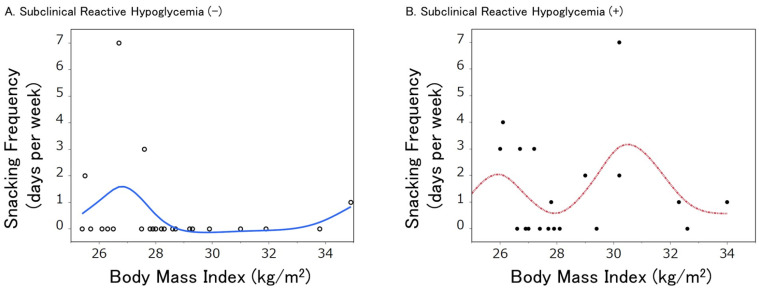
Snacking frequency in obese/overweight men as a function of body mass index (BMI) according to the absence (**A**) or presence (**B**) of subclinical reactive hypoglycemia (SRH).

**Figure 2 metabolites-13-00754-f002:**
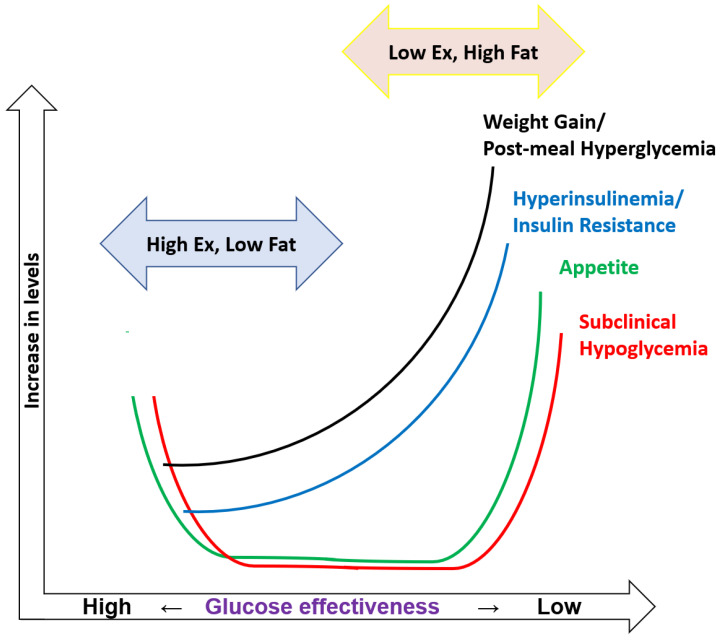
A conceptual diagram of the role of subclinical hypoglycemia-induced appetite depending on the levels of glucose effectiveness.

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
