# Peer review of "Subclinical Reactive Hypoglycemia with Low Glucose Effectiveness—Why We Cannot Stop Snacking despite Gaining Weight"

_metabolites, 2023, doi:10.3390/metabo13060754_

Round 1
Reviewer 1 Report
This manuscript describes the impact of snacking habits on glucose effectiveness and reactive hypoglycemia/hyperglycemia. The manuscript is well written and well organised. The manuscript lacks depth and interest for the general audience. Although the author state the possibility that hypoglycaemia may induce uncontrollable hunger, the explanation is very superficial. It is known that a postprandial inflammatory response (PPIR) affects glucose levels (up and down) and even the role of brown adipose tissue in postprandial and continuous glucose levels are not mentioned. The possibility of uncontrollable hunger can also be caused by an acute or chronic neuroinflammation of the hypothalamus. The solutions based on certain pharmacological interventions are also very superficial. First no contra-indications are mentioned. Second, the impact of low grade inflammation and PPIR on glucose levels is not mentioned anywhere in this manuscript and as possible treatment targets. Pharmacological interventions will produce negative effects on the long run anyway and prevention with medicines is not really recommendable. To publish this paper more detailled information is needed and treatment conclusions should be more general and actualised (see impact of brown adipose tissue on glucose clearance capacity)
English only needs some corrections.
Sentences 78-83 have to be rewritten.
Author Response
To Reviewer 1,
Thank you for your instructive comments. I appreciate your time and effort for reviewing the paper as well as your helpful suggestions. To add depth and interest for the general audience, the following modifications were included in the newly revised manuscript.
Q1. It is known that a postprandial inflammatory response (PPIR) affects glucose levels …
A1. The paragraph concerning PPIR was included as shown below:
In lines 367-385,
“13. Postprandial inflammatory response and Sg
It is known that ingestion of a high-fat diet, high-carbohydrate diet, or combination causes postprandial responses associated with activation of the innate immune system, which initiates systemic low-grade inflammation [66]. High-fat meal consumption results in a temporary pro-inflammatory state via “metabolic endotoxemia”, in which bacterial wall products derived from gut microbiota such as lipopolysaccharide (LPS) can be found in the blood circulation [67]. In healthy volunteers, endotoxin challenge (1 ng/kg LPS) significantly decreased Sg derived from a frequently sampled intravenous glucose tolerance test [68]. Therefore, increases in plasma LPS, and pro-inflammatory cytokines such as interleukin-6 (IL-6) or tumor necrosis factor-α (TNF-α) levels could lead not only to insulin resistance [67] but also to decreased Sg [68], both of which may result in postprandial hyperglycemia, and following reactive hypoglycemia.
In mice models with gut microbiota depletion, it is reported that glucose uptake in BAT is enhanced [69]. A local pro-inflammatory environment in BAT alters the thermogenic activity by impairing its energy expenditure mechanism and glucose uptake for use as a fuel substrate [70]. Therefore, the ”metabolic endotoxemia” might lead to the inhibition of glucose uptake by BAT, which results in lower energy expenditure as well as postprandial hyperglycemia, especially when the amount of active BAT is abundant, such as in children and adolescents or in adults with cold exposure.”
Q2. …the role of brown adipose tissue in postprandial and continuous glucose levels ..
A2. The paragraph regarding brown adipose tissue was included as shown below:
In lines 353-366,
“12. Brown adipose tissue (BAT) and Sg
In healthy women treated with a β3-adrenergic receptor agonist, it is reported that the metabolic activity of BAT measured by [18F]-2-fluoro-d-2-deoxy-d-glucose (18F-FDG) PET/CT is increased, which is accompanied by higher Sg as well as insulin sensitivity, and insulin secretion [63], suggesting that glucose uptake by BAT may in-crease Sg.
Cold exposure is one of the best-explored activators of brown adipocytes. During the ice age, the latest of which peaked about 20,000 years ago, it is speculated that global temperatures were likely about 10°F (5°C) colder than today [64]. Global warming as well as heating equipment such as air-conditioning in civilized countries may lead to the de-activation of brown adipocytes, which could also be involved in the tendency towards low Sg in modern lives. Since BAT increases not only energy expenditure but also glucose clearance [65], earlier loss of brown adipocytes could lead to decreased postprandial glucose disposal and, thus, to hyperglycemia after meals.”
Q3. The possibility of uncontrollable hunger can also be caused by acute or chronic neuroinflammation of the hypothalamus.
A3. The paragraph regarding neuroinflammation was included as shown below:
In lines 386-402,
“14. Reactive hypoglycemia and inflammation
Previous studies documented that, in both diabetes and non-diabetes, markers of inflammation such as C-reactive protein, and IL-6 are upregulated after acute hypoglycemia [71-73]. Combined with endothelial dysfunction, hypercoagulability, and activation of the sympathetic nervous system, the chronic inflammatory mechanism is thought to be involved in the pathogenesis of atherosclerotic diseases in postprandial reactive hypoglycemia [74,75].
It is reported that several pro-inflammatory cytokines such as TNF-α, interleukin (IL)-6, IL-1α, and IL-1β can cross the blood-brain barrier (BBB), while receptors for these proinflammatory cytokines are expressed in the hypothalamus [76]. The pro-inflammatory cytokines-induced low-grade inflammation changes the levels of neuropeptides involved in the hypothalamic and mesolimbic appetite-controlling systems, which could be associated with overeating through non-homeostatic (hedonic) pathways [77]. Along with nutritional signals, such as saturated fatty acids, which have been shown to cause chronic hypothalamic inflammation, an increase in food intake, and weight gain [78], chronic low-grade inflammation caused by SRH might also contribute to excess food intake.”
Q4. The solutions based on certain pharmacological interventions are also very superficial.
First no contra-indications are mentioned. Second, the impact of low-grade inflammation and PPIR on glucose levels is not mentioned anywhere in this manuscript and as possible treatment targets.
A4. The sentences regarding pharmacological intervention were included as shown below:
In lines 330-331,
“..., although beneficial effects of any interventions should be balanced with their adverse reactions and costs.”
In lines 351-352,
“In addition, as shown in the following paragraphs, brown adipocyte and low-grade inflammation could also be potential therapeutic targets in the future.”
.
Q5. Sentences 78-83 have to be rewritten.
A5. The sentences were rewritten as shown below:
In lines 79-85,
“Although, in theory, regular ingestion of snacks in large portions to exceed the body’s daily energy needs could contribute to weight gain, as of today, researchers have failed to present a definite positive causal relationship between snacking frequency and long-term weight gain in a prospective study [16]. The results of the previous epidemiological studies indicate that excess of calories with frequent snacking may be associated with but not sufficient for developing obesity, where additional factor(s) may also be involved.”

Reviewer 2 Report
The review article by Ichiro Kishimoto covers an important aspect of increasing obesity in current population. The review is focussed and follows logically to elaborate the interplay of SRH and Sg in causing frequent snacking. There are few suggestions to improve the readability-
1. Since continuous glucose monitoring appears frequently, few lines on explaining how it is done will benefit the readers.
2. I find fitting of splines to categorical data not much useful. The author can explore other statistical ways to capture the idea instead of fitting continuous functions like splines.
3. Please indicate which level in the y-axis of figure 2. "Increase in levels" is not self explanatory.
Author Response
To Reviewer 2,
Thank you for your instructive comments. I appreciate your time and effort for reviewing the paper as well as your helpful suggestions.
Q1. Since continuous glucose monitoring appears frequently, few lines on explaining how it is done will benefit the readers.
A1. Few lines on explaining CGM were added in the newly revised manuscript as shown below:
Lines 107-109,
“Continuous glucose monitoring (CGM) is wearable technology, which, through a tiny sensor inserted subcutaneously, automatically tracks blood glucose levels continuously 24 hours a day.“
Q2. I find fitting of splines to categorical data not much useful. The author can explore other statistical ways to capture the idea instead of fitting continuous functions like splines.
A2. In the revised manuscript, Figure 1 was replaced where BMI (kg/m2) was plotted against snacking frequency (continuous variable) instead of snacking habits (categorical variable).
Sentences describing the figure were also added as shown below:
In lines 156-160,
“When the relationships between snacking frequencies (snacking days per week) and BMI (kg/m2) were fit with spline curves according to the presence or the absence of SRH, less snacking is observed by increasing BMI in those without SRH, while, in subjects with SRH, obese people still snack regularly even if their BMI is high (Figure 1).”
In lines 169-176,
“Figure 1. Snacking frequency in obese/overweight men as a function of body mass index (BMI) according to the absence (A) or the presence (B) of subclinical reactive hypoglycemia (SRH).
Plots of BMI (kg/m2) against snacking frequency (days per week) were created for 43 males from data previously reported [7]. Data were further assessed by the cubic spline functions and smooth curve fitting with lambda=1 (the solid line in A and the dashed line in B). SRH was defined as previously reported [7]. Snacking frequencies were self-reported. BMI was defined as body weight (kg) divided by height (m) squared.”
Q3. Please indicate which level in the y-axis of figure 2. "Increase in levels" is not self explanatory.
A3. Explanatory sentences for the y-axis were also added as shown below:
In lines 270-272,
“The vertical axis shows the likelihood of the respective parameters (i.e. weight gain, post-meal hyperglycemia, hyperinsulinemia, insulin resistance, increase in appetite and SRH), while the horizontal axis shows the level of Sg.”

Round 2
Reviewer 1 Report
Super